# Production of Multifunctional Hydrolysates from the *Lupinus mutabilis* Protein Using a *Micrococcus* sp. PC7 Protease

**DOI:** 10.3390/biotech14020032

**Published:** 2025-04-27

**Authors:** Keyla Sofía Llontop-Bernabé, Arturo Intiquilla, Carlos Ramirez-Veliz, Marco Santos, Karim Jiménez-Aliaga, Amparo Iris Zavaleta, Samuel Paterson, Blanca Hernández-Ledesma

**Affiliations:** 1Laboratory of Molecular Biology, Faculty of Pharmacy and Biochemistry, Universidad Nacional Mayor de San Marcos, Lima 01, Peru; keyla.llontop@unmsm.edu.pe (K.S.L.-B.); carlos.ramirez29@unmsm.edu.pe (C.R.-V.); kjimeneza@unmsm.edu.pe (K.J.-A.); azavaletap@unmsm.edu.pe (A.I.Z.); 2Department of Bioactivity and Food Analysis, Institute of Food Science Research (CIAL, CSIC-UAM, CEI-UAM + CSIC), Nicolás Cabrera 9, 28049 Madrid, Spain; samuel.paterson@csic.es

**Keywords:** *Lupinus mutabilis*, albumin fraction, ultrafiltration, protease PC7, multifunctionality

## Abstract

The growing demand for functional foods has driven the search for bioactive compounds derived from plant proteins. *Lupinus mutabilis* “Tarwi”, a legume native to the Peruvian Andes, stands out for its high protein content and potential as a source of bioactive peptides (BPs). In this study, the functionality of the proteins contained in the albumin fraction (AF) isolated by tangential ultrafiltration (TFF) was investigated by using the OmicsBox software. The identified proteins were functionally classified into three groups: cellular component (35.57%), molecular function (33.45%), and biological process (30.97%). The isolated AF was hydrolysed with the native protease PC7 (HAP), optimizing the E/S ratio and time parameters. Additionally, sequential hydrolysis of the PC7 protease and alcalase (HAPA) was performed. In vitro multifunctionality assays, HAP and HAPA demonstrated the ability to scavenge radicals (ABTS and ORAC) and inhibit angiotensin-converting enzyme (ACE)-I and dipeptidyl peptidase IV (DPP-IV). The findings of this study highlight the potential of *L. mutabilis* albumin hydrolysate as a multifunctional ingredient for functional foods aimed at managing chronic conditions associated with oxidative stress, hypertension, and/or metabolic disorders.

## 1. Introduction

In recent years, changes in human diet behaviour have boosted the consumption of functional foods, with an estimated annual growth of 8.5% until 2030 [1]. Plant-based proteins offer an alternative source of bioactive compounds, presenting advantages over animal-based proteins due to their diversity, cost-effectiveness, large-scale production, and environmental sustainability [2]. Legumes stand out among plant sources for their nutritional value, as they constitute a source of high-quality proteins, amino acids (AAs), and minerals [3]. Beyond their nutritional contribution, legumes contain a variety of bioactive compounds, such as isoflavones, bioactive peptides (BPs), saponins, phenolic compounds, galactosides, and anti-nutritional factors [4,5]. While the latter, such as phytic acid and alkaloids, have traditionally been associated with reduced nutrient bioavailability, emerging evidence highlights their potential bioactive properties, including antioxidant and antidiabetic effects [6,7]. These bioactive components exhibit metabolic, hormonal, and digestive regulatory effects, in addition to prebiotic properties that vary depending on the specific compound type, as well as digestibility and bioavailability [8,9,10].

BPs are short sequences of AAs released from the native protein through in vitro enzymatic hydrolysis, fermentation, and/or gastrointestinal digestion. BPs primarily can exhibit antioxidant, antihypertensive, and antidiabetic properties [11]. Their biological activity is determined by the AA sequence, the type of terminal AA, the length of the chain, the electrical charge, and the hydrophobic and hydrophilic properties, among others [12]. In particular, the aromatic (tryptophan, tyrosine, and phenylalanine), imidazole ring (histidine), and sulphur (cysteine and methionine) AAs present in BPs have been recognized to contribute to their ability to donate hydrogen or electrons, inhibiting the formation of free radicals [9,13]. This mechanism is crucial for the prevention of metabolic and chronic diseases associated with oxidative stress, such as obesity, type II diabetes, neurodegenerative disorders, and cancer [12].

Studies have reported that legume proteins such as *Lupinus albus* and *L. angustifolius* exhibit antioxidant, antihypertensive, and antidiabetic activity primarily after being treated with commercial enzymes such as pepsin, pancreatin, flavourzyme, neutrase, and alcalase [14,15,16,17]. Although native proteases from non-commercial bacterial sources have been investigated for their ability to release antioxidant peptides [18], their application to legume proteins has not yet been explored. In this regard, the PC7 protease from *Micrococcus* sp., isolated from *Pilluana salterns* in Peru, represents a promising biocatalyst for generating unique BPs from Andean legumes, as demonstrated in the study by Bautista et al. [19]. This protease operates at pHs (7.0 to 8.0) and temperatures (25 to 40 °C) that avoid adverse handling conditions, favouring its scaling. This prevents excessive protein denaturation, preserving the functional properties of the released peptides. To enhance peptide yield and reduce hydrolysis time and process cost, it is essential to optimize the hydrolysis conditions based on the parameters of temperature, pH, reaction time, and enzyme concentration.

*L. mutabilis* “Tarwi”, a legume native to the Peruvian Andes, is notable for its high protein content, ranging from 32.0 to 52.6% (*w*/*w*), primarily composed of albumin and globulin fractions. Traditionally, lupin storage proteins are also known as α-, β-, γ-, and δ-conglutins [20]. The growing interest in these proteins is not only due to nutritional reasons but also because they are a source of bioactive compounds. A total of 25 peptides with antioxidant, antihypertensive, and antidiabetic capacities have been identified in a protein hydrolysate of *L. mutabilis*; however, the exact precursor protein is unknown [21]. The albumin fraction is one of the proteins with the greatest potential for industrial development due to its high solubility in pure water, which favours its extraction and processing, making it a valuable ingredient for the formulation of healthy products [20,22]. The quality and functionality of these proteins depend to a large extent on the extraction method used. In this sense, tangential flow ultrafiltration (TFF) is one of the most efficient techniques, as it allows soluble proteins to be isolated in their native state while preserving their structure and functionality [23].

Proteomic analysis based on high-resolution mass spectrometry is a powerful tool for optimizing protein characterisation. This technique allows the identification, quantification, and analysis of proteins, even those found at low concentrations, allowing the study of their main functions in plants [24,25]. Previous proteomic studies on *Lupinus* spp. have focused on taxonomic identification, allergen detection, and conglutin diversity characterization [24,25,26]. This highlights the importance of expanding proteomic research on the albumin of *L. mutabilis* to explore its potential as a source of BPs for functional food formulations.

In this context, this study aimed to harness the albumin fraction (AF) proteins of *L. mutabilis*—characterized for the first time via proteomic analysis—to develop a multifunctional protein hydrolysate with in vitro antioxidant, antihypertensive, and antidiabetic properties. To achieve this, we employed the novel native PC7 protease (HAP), previously undescribed for this application. Furthermore, sequential hydrolysis using PC7 protease followed by alcalase (HAPA) significantly enhanced these bioactive properties. Beyond advancing scientific understanding, this innovative approach enhances the nutritional and functional value of *L. mutabilis* seeds while promoting their economic valorisation and the creation of novel food ingredients.

## 2. Materials and Methods

### 2.1. Materials and Reagents

The seeds of *L. mutabilis* “Tarwi” were collected in Huamachuco (Sanchez Carrion- La Libertad, Peru) in 2023.

Alcalase 2.4 L (*Bacillus licheniformis*, >2.4 U/g) was obtained from Sigma-Aldrich (Bagsvaerd, Denmark), 2,2′-azino-bis (3-ethylbenzothiazoline-6-sulfonic acid) (ABTS), 6-hydroxy-2,5,7,8-tetramethylcroman-2-carboxylic acid (Trolox), fluorescein disodium (FL), 2,2′-azobis dihydrochloride (2-amidinopropane) (AAPH), and the dipeptidyl peptidase IV (DPP-IV) assay kits were obtained from Cayman Chemicals (Ann Arbor, MI, USA), and orthophthalaldehyde (OPA) was obtained from Merck Millipore Corp (Darmstadt, Germany). All other chemicals were of analytical grade.

### 2.2. Production of PC7 Protease

The strain *Micrococcus* sp. PC7 was provided by the Molecular Biology Laboratory, Faculty of Pharmacy and Biochemistry, Universidad Nacional Mayor de San Marcos (Lima, Peru). The culture was grown in a medium containing 10 g/L glucose, 5 g/L yeast extract, and 5% salt water, adjusted at pH 8.0 [27]. The strain was inoculated into the medium at an optical density (OD)_600_ of 0.5, using 10% (*v*/*v*) of the culture in 400 mL of medium. Incubation was carried out at 30 °C, 150 rpm, for 48 h. After incubation, cells were removed by centrifugation at 8000× *g* for 20 min, and the resulting supernatant was collected as the source of proteases, referred to as the crude extract. This crude extract was concentrated to a final volume of 10 mL using a 10 kDa molecular weight cut-off (MWCO) membrane (Millipore, CA, USA), at 7500× *g* for 20 min at 4 °C (Bautista et al., 2024 [19]).

The proteolytic activity of the crude extract was determined following the method described by Bautista et al. [19]. One unit of enzyme activity (U) was defined as the amount of enzyme required to produce a 0.1 increase in absorbance at 280 nm, under the specified test conditions. Specific activity was expressed as U/mg of protein.

### 2.3. Obtention of Albumin Fraction from L. mutabilis Seeds

The albumin fraction (AF) was extracted from *L. mutabilis* seeds following the method described by Sironi et al., with minor modifications [28]. Defatted flour was dispersed in Milli-Q water 1:9 (*w*/*v*), homogenized, and adjusted to pH 7.0 using 1 M NaOH. The suspension was stirred with a magnetic stirrer (VWR 4X4, PA, USA) at 1000 rpm for 1 h at 24 °C. Subsequently, the suspension was centrifuged at 2700× *g* for 25 min at 10 °C and the supernatant was collected.

The protein extract underwent TFF using a Pellicon^®^ 3 cassette system with an Ultracel^®^ regenerated cellulose membrane (nominal MWCO of 3 kDa, total area 0.11 m^2^; Millipore corporation, Billerica, MA, USA). The process was performed at an initial transmembrane pressure (TMP) of 24 psi with a feed flow rate of 450 mL/min. The TFF continued until the volume was reduced to 200 mL, achieving a volumetric concentration factor (VCF) of 5.75. Both retentate and permeate samples were collected for subsequent analyses. The retentate, representing the concentrated AF, was freeze-dried, and stored at 4 °C until further use. The protein content was determined using the bicinchoninic acid (BCA) method, following the instructions of the Pierce™ BCA Protein Assay Kit (Thermo Fisher Scientific, Waltham, MA, USA). Bovine serum albumin (BSA, 25–1000 μg/mL) was used as the standard.

### 2.4. Characterization of the Albumin Fraction

#### 2.4.1. Determination of Protein and Amino Acid Content

The protein concentration of AF was also measured via the Kjeldahl method [29], employing a block digester (J.P. Selecta, Barcelona, Spain) and a Buchi Kjeldahl K-314 distillation unit (BÜCHI Labortechnik AG, Flawil, Switzerland). The analysis was carried out in duplicate using a conversion factor of 5.71. Additionally, the amino acid content was measured in duplicate using a Biochrom 30 Series Amino Acid Analyser (Biochrom, Cambridge, UK), following the previously internationally established methodology [30].

#### 2.4.2. In-Gel Digestion (Stacking Gel)

The AF was suspended in 50 µL of sample buffer and loaded into 1.2 cm-wide wells of a standard SDS-PAGE gel (0.75 mm thick, 4% stacking, 10% resolving; Bio-Rad, Hercules, CA, USA) following a previously described protocol [31]. The run was halted once the front migrated 3 mm into the resolving gel, concentrating the entire proteome at the stacking/resolving gel interface. Unseparated protein bands were stained with Coomassie dye (Bio-Rad), excised, cut into 2 × 2 mm cubes, and transferred to 0.5 mL microcentrifuge tubes [32]. The gel fragments were destained using an ACN/H_2_O (1:1, *v*/*v*) solution, then subjected to reduction and alkylation—cysteinyl disulfide bonds were reduced with 10 mM 1,4-dithiothreitol (DTT) at 56 °C for 1 h, followed by alkylation with 10 mM iodoacetamide at room temperature for 30 min in the dark. In situ digestion was carried out using sequencing-grade trypsin (Promega, Madison, WI, USA) as per Shevchenko et al. [33]. The gel pieces were dehydrated with acetonitrile (ACN), which was subsequently removed, and then dried in a speedvac before being rehydrated in 100 mM Tris-HCl (pH 8.0), 10 mM CaCl_2_ containing 60 ng/mL trypsin at a 5:1 protein/enzyme (*w*/*w*) ratio. The samples were kept on ice for 2 h and incubated at 37 °C for 12 h. Digestion was terminated with the addition of 1% trifluoroacetic acid (TFA). The resulting supernatants were dried and desalted using OMIX Pipette Tips C18 (Agilent Technologies, Santa Clara, CA, USA) before analysis. The digestion process was carried out in the presence of 0.2% RapiGest (Waters, Milford, MA, USA).

#### 2.4.3. Reverse-Phase Liquid Chromatography (RP-LC-MS/MS) Analysis (Dynamic Exclusion Mode)

The desalted protein digest was dried, reconstituted in 10 µL of 0.1% formic acid, and analysed using RP-LC-MS/MS on an Easy-nLC 1200 system coupled to an LTQ-Orbitrap Velos Pro hybrid mass spectrometer (Thermo Scientific, Waltham, MA, USA). Peptides were first concentrated online via reverse-phase chromatography with a 0.1 mm × 20 mm C18 RP precolumn (Thermo Scientific) and then separated on a 0.075 mm × 250 mm bioZen 2.6 µm Peptide XB-C18 RP column (Phenomenex, Torrance, CA, USA) at a flow rate of 0.25 μL/min. Elution was carried out over 180 min using a dual-gradient system: 5–25% solvent B over 135 min, 25–40% over 45 min, 40–100% over 2 min, followed by 100% solvent B for 18 min (solvent A: 0.1% formic acid in water; solvent B: 0.1% formic acid, 80% ACN in water). Electrospray ionization (ESI) was performed using a Nano-bore stainless steel emitter (30 μm ID; Proxeon, Odense, Denmark) with a spray voltage of 2.1 kV and an S-Lens setting of 60%. The Orbitrap resolution was set at 30,000 [34]. Peptides were detected in survey scans ranging from 400 to 1600 amu (1 μscan), followed by twenty data-dependent MS/MS scans (Top 20) with a 2 u isolation width (*m*/*z* units), a normalized collision energy of 35%, and dynamic exclusion for 60 s. Charge-state screening was enabled to discard unassigned and singly charged protonated ions. The mass spectrometry proteomics data have been deposited to the ProteomeXchange Consortium via the PRIDE [35] partner repository with the dataset identifier PXD062562 and 10.6019/PXD062562.

#### 2.4.4. Data Processing

Peptide identification from raw data was conducted using the PEAKS Studio v11.5 search engine (Bioinformatics Solutions Inc., Waterloo, ON, Canada). The database search was performed against uniprot-phaseoleae.fasta (480,563 entries; UniProt release 02/2024) or a combined database consisting of uniprot-lupinus-albus, uniprot-lupinus-angustifolius, and uniprot-lupinus-mutabilis.fasta (69,007 entries; UniProt release 06/2024) using a decoy-fusion database. Search parameters included semi-specific tryptic cleavage at arginine and lysine, allowance for up to two missed cleavage sites, and mass tolerances of 20 ppm for precursor ions and 0.6 Da for MS/MS fragment ions. Modifications considered in the search included optional methionine oxidation and cysteine carbamidomethylation. The false discovery rate (FDR) was restricted to 0.01 for both peptide–spectrum matches (PSMs) and protein identifications. Proteins were considered confidently identified only if at least two unique peptides were detected in LC-MS/MS analyses [34,36].

#### 2.4.5. Proteomic Functional Analysis

A proteomic functional analysis of all the detected proteins from AF was carried out through the BLAST2GO methodology [37] using the OmicsBox 3.4 bioinformatic software (Biobam, Valencia, Spain). Three distinct functional groups were made: cellular component, molecular function, and biological process. To complete the alignment, the non-redundant database for protein sequences was used using the taxonomy filter: *Phaseoleae* (code 163735), *L. albus* (code 3870), *L. angustifolius* (code 3871), and *L. mutabilis* (code 53232). The step by step full workflow followed for the functional analysis is shown in Appendix A.

### 2.5. Hydrolysis of Albumin Fraction with PC7 Protease

An aqueous suspension of the AF (65 mg/mL) was prepared and heated to 85 °C for 10 min before hydrolysis with PC7 protease. The reaction conditions were maintained at pH 8.0, a temperature of 40 °C, and 150 rpm agitation [19]. The E/S ratio and hydrolysis time (T) were defined using a central composite design (CCD). The E/S ratio ranged from 44 to 256 U/mg of substrate (X_1_), while the hydrolysis time ranged from 26.4 to 153.6 min (X_2_). The dependent variables were the degree of hydrolysis (*DH*) and the antioxidant capacity (ABTS). A total of 11 assays were performed, based on five levels of the independent variables, including 4 factorials, 4 axial points, and 3 replicates at the central point. The experimental design was generated using the Statistica v10.0 software.

The regression models describing the relationship between the dependent variables (Y) and independent variables (X) were expressed using the following polynomial equation:(1)$$Y=\beta_{0}+\beta_{1}X_{1}+\beta_{2}X_{2}+\beta_{3}X_{1}^{2}+\beta_{4}X_{2}^{2}+\beta_{5}X_{1}X_{2}$$

Enzymatic hydrolysis was terminated by heating the reaction at 100 °C for 10 min, followed by cooling and centrifugation at 5000× *g* for 10 min at 4 °C. The resulting supernatants were freeze-dried and stored at −20 °C for subsequent use.

The *DH* was determined based on the method described by Nielsen et al. by measuring the free amino groups (*h*) by reaction with OPA and expressed as mg L-serine/mL [38]. *DH* (%) was calculated using the formula:(2)$$\%DH=\frac{h}{ht}\times 100$$

The total amino group content (*ht* = 10.15) was determined in samples subject to complete hydrolysis (100%) with 6 N HCl at 121 °C for 1 h. Soluble protein concentrations in the hydrolysates were quantified using the BCA assay as outlined in Section 2.3.

### 2.6. Sequential Hydrolysis of Albumin Fraction with Protease PC7 and Alcalase (HAPA)

The AF was sequentially hydrolysed using PC7 protease at a ratio of 168 U/mg substrate, under conditions of pH 8.0 and 40 °C, for 105 min at 150 rpm (HAP). This was followed by the hydrolysis with alcalase at an E/S ratio of 1:50 (*w*/*w*), pH 8.5, and 50 °C for 2 h [39]. The reaction was terminated by heating at 100 °C for 10 min, followed by centrifugation at 5000× *g* for 10 min at 4 °C. The resulting supernatant (HAPA) was freeze-dried and stored at −20 °C for subsequent use.

### 2.7. SDS-PAGE Electrophoretic Profile

The electrophoretic profiles of AF, HAP, and HAPA were analysed using SDS-PAGE-Tricine, under reducing and non-reducing conditions, following the method described by Haider et al., with slight modifications [40]. Samples were treated with sample buffer containing 1% (*w*/*v*) sodium dodecyl sulphate (SDS), 24% glycerol, and 0.02% (*w*/*v*) Coomassie brilliant Blue in 0.1 mM Tris-HCl buffer at pH 6.8. For reducing conditions, 4% (*v*/*v*) 2-mercaptoethanol was added. The samples were incubated at 65 °C for 10 min. Protein aliquots (10 μg) were loaded onto a resolving gel (10% acrylamide) with a stacking gel (4% acrylamide). Electrophoresis was conducted in a vertical gel unit (Mini PROTEAN, Bio-Rad) at 80 V for 20 min, followed by 100 V for 2 h. An MW marker ranging from 10 to 225 kDa (Perfect Protein Marker, Sigma-Aldrich) was used.

### 2.8. Determination of Biological Activities

#### 2.8.1. ABTS Assay

The ABTS^•+^ radical scavenging activity was measured according to the method of Re et al., with minor modifications [41]. The 7 mM ABTS^•+^ radical was generated by mixing 3 mL of 10 mM ABTS with 3 mL of 2.45 mM potassium persulfate. The reaction mixture was protected from light by storing it in an amber glass vial and was allowed to stand for 16 h at 23 °C. Subsequently, the solution was diluted with 5 mM PBS (pH 7.4) to achieve an absorbance of 0.70 ± 0.02 at 734 nm. Trolox was used as a standard at concentrations ranging from 2 to 25 μM. For the assay, 180 µL of ABTS^•+^ solution was mixed with 20 μL of PBS (blank), Trolox (standard), or sample (hydrolysate), and the absorbance was measured at 734 nm after 5 min incubation at room temperature, using a Tecan Infinite 200PRO plate reader (Tecan Group AG, Männendorf, Switzerland), controlled by the Icontrol software version 1.11.10. The results were expressed as µmol of Trolox equivalents (TEs) by g of protein. Samples were analysed in triplicate.

#### 2.8.2. Oxygen Radical Absorbance Capacity (ORAC) Assay

The ORAC assay was performed as described by Hernández-Ledesma et al., with slight modifications [42]. Briefly, a 200 μL reaction mixture containing disodium fluorescein (116.61 nM), 14 mM AAPH, and either Trolox (0–3 nM) or sample (hydrolysate) was prepared in 75 mM PBS (pH 7.4). The mixture was incubated at 37 °C, and fluorescence was measured at the excitation wavelength of 485 nm and emission wavelength of 520 nm every 2 min over 120 min using a plate reader Tecan Infinite 200PRO (Tecan Group AG). The values were expressed as μmol TE/mg of protein. Samples were analysed in triplicate.

#### 2.8.3. Angiotensin-Converting Enzyme (ACE) Inhibitory Activity

The angiotensin-converting enzyme (ACE) inhibitory activity was determined using the methodology described by Hayakari et al., with slight modifications [43]. Briefly, a 200 µL mixture containing 1 mU ACE, 1.25 mM hippuryl histidyl-leucine, 0.05 µM Captopril (control), or sample (hydrolysate) was prepared and incubated at 37 °C for 60 min. The reaction was terminated by heating the samples at 100 °C for 1 min. Subsequently, 100 μL of a 3% (*v*/*v*) trichloro-5-triazine/dioxane solution was added, and the mixture was stirred and centrifuged at 1000× *g* for 10 min. Finally, absorbance was measured at 382 nm using the Tecan Infinite 200PRO (Tecan Group AG) plate reader. The results were expressed as sample concentration required to inhibit ACE activity by 50% (IC_50_). Samples were analysed in triplicate.

#### 2.8.4. Dipeptidyl Peptidase-IV (DPP-IV) Inhibitory Activity

The antidiabetic activity of the hydrolysates was assessed by evaluating their DPP-IV inhibitory activity [44]. A 100 μL reaction mixture was prepared, consisting of 5 μL of enzyme solution, 5 μL of substrate, sitagliptin (standard), or samples at various concentrations. The mixture was incubated at 37 °C for 30 min. Fluorescence measurements were recorded every 2 min (λ excitation = 360 nm/λ emission = 460 nm) using the Tecan Infinite 200PRO (Tecan Group AG) plate reader. The results were expressed as the sample concentration required to inhibit 50% of DPP-IV activity (IC_50_). Samples were analysed in triplicate.

### 2.9. Statistical Analysis

All experiments were performed in triplicate and presented as mean ± standard deviation. The data were subjected to a one-way analysis of variance (one-way ANOVA) followed by Tukey’s test. Differences were considered statistically significant when *p* < 0.05. Figures were prepared using GraphPad Prism 8 software. Analysis and interpretation of data on hydrolysis optimization were performed using Statistica v10.0 software.

## 3. Results and Discussion

### 3.1. Obtention of Albumin Fraction (AF) from Lupinus mutabilis by Tangential Flow Ultrafiltration

The AF is a water-soluble, sulphur-rich protein (δ-conglutin) that performs storage and regulatory functions in seeds [45]. TFF was employed as an alternative to isoelectric precipitation for AF isolation. This technique facilitates the fractionation of the target protein while preserving its native structure and functionality [46]. The work design is shown in Figure 1A and the operating parameters are given in Appendix A.

The volumetric flow rate remained relatively stable throughout the 40 min process. This stability was counterbalanced by a slight increase in the TMP, which rose from 24.0 ± 0.35 to 26.13 ± 1.94 psi, without a statistically significant reduction (*p* < 0.05) in the normalized water permeability (NWP) parameter. However, a total protein loss of 16.48 ± 0.49% was detected (Figure 1B). This loss could be attributed to permeate entrainment phenomena and protein–membrane interactions, both of which contribute to protein loss and are linked to the fouling index. Membrane fouling is shown in Figure 1C. No significant accumulation of fouling material was detected on the membrane until 30 min after the process began. Subsequently, a slight increase in fouling was observed, reaching 3.03 ± 1.43% after 40 min. This trend might be explained by the increase in protein concentration, which promotes concentration polarization on the membrane surface and partial pore blockage as the process progresses [47]. These findings underscore the importance of optimizing the UF process by carefully selecting membranes and controlling operating parameters to achieve desired yields while minimizing fouling and polarization effects.

The TFF approach enabled a 4.36-fold concentration of albumin protein (Figure 1B), achieving a VCF of 5.75, and yielding a 2.9-fold (75.82% *w*/*w*) increase in recovery compared to the isoelectric precipitation process (26.23%) reported in a previous study [15]. The improvement of the yield (%) could be attributed to the ability of TFF to separate and concentrate proteins in a controlled environment, minimizing protein loss during the process. Similarly, Hojilla et al. demonstrated the efficacy of UF combined with diafiltration (DF), achieving a 72% recovery rate in *L. albus* using a 5 kDa polyether sulfone membrane, with a flow rate of 2.7 L/h [48]. These findings align with the results obtained in the current study, reinforcing the notion that membrane-based technologies are versatile and effective tools for processing plant-derived proteins.

The protein content of the resultant AF was 82.87 ± 0.21%. Its AA composition is shown in Table 1. The chromatogram obtained from the analysis is shown in Appendix A.

The amino acid Trp was not detected according to the official AOAC International method [30]. This process was performed with 6 N HCl at high temperatures (110 °C) for 24 h to break peptide bonds. The indole ring of Trp is highly susceptible to degradation under acidic conditions, forming undetectable non-volatile compounds in the final hydrolysate. Among the essential AAs (EAAs), Leu and Lys were the most abundant, with values of 3.73 ± 0.02 and 3.52 ± 0.05 g/100 g of protein, respectively. Asp + Asn, Arg, and Glu + Gln were the most abundant non-essential AAs (NEAAs), with values of 20.01 ± 0.26, 8.54 ± 0.29, and 6.41 ± 0.03 g/100 g of protein, respectively. The ratio of EAAs to total AAs (TAAs) was 29%, and the ratio of EAAs to NEAAs was 40%. Both values were lower than the protein reference pattern (EAAs/TAAs, 40%; EAAs/NEAAs, 60%) raised by FAO/WHO.

Figure 2 illustrates the electrophoretic profile under reducing (β-mercaptoethanol) and non-reducing conditions of FA obtained by TFF. Under reducing conditions, bands with estimated molecular weights of 10, 14, 20, and 24 kDa were observed. Under non-reducing conditions, protein bands of 16 and 18 kDa were additionally detected. These findings were consistent with those reported by Nadal et al. for *L. luteus*, where bands around 20 kDa and below 15 kDa were observed [49]. Similarly, Muranyi et al. identified an albumin with an MW of approximately 13 kDa in *L. angustifolius* seeds [50]. Salmanowicz further noted that the AF consisted of polypeptide bands containing four to eight isoforms, characterized by two subunits of 3–6 kDa and 8–12 kDa, linked by four disulfide bridges, with variations depending on the species [51,52]. These observations were supported by Foley et al., who highlighted genetic variability within the same species, influenced by the seed development stage [53].

### 3.2. Proteomic Characterization and Functional Analysis of L. mutabilis Albumin

Following RP-LC-MS/MS analysis of the AF sample and data processing using the PEAKS and SPIDER search tools [54], a total of 2424 peptide–spectrum matches (PSMs) and 2474 scans (representing MS/MS scans associated with peptide spectra) were identified. Additionally, 955 features were detected through database searches alone, 887 peptide sequences contained modifications excluding isoleucine/leucine differentiation, and 564 features were identified without modifications or isoleucine/leucine differentiation. Furthermore, the PEAKS software detected a total of 242 proteins following the database search and RP-LC-MS/MS analysis. Since homology-based protein identification relies on sequence data, PEAKS organizes proteins into groups that include all proteins identified with the same peptides [31,55]. In our case, 111 protein groups were identified, and, out of the 242 proteins detected, 168 proteins were reliably identified with two or more unique peptides, and 74 proteins had less than two unique peptides.

The proteomic features and details of proteins with at least two unique peptides detected in the AF sample, were collected and are made available in Appendix A. The AF constitutes approximately 10% of the total protein content in *L. mutabilis* seeds. Within this fraction, while some proteins, such as δ-conglutin, are well-characterized components, others have not yet been comprehensively described in the available literature [56]. Therefore, further proteomic studies would be necessary to identify and quantify the specific proteins present in this fraction. Concerning the proteomic functional analysis, all 242 proteins detected in the sample were used. This strategy made it possible to detect proteins that might be missed or underrepresented due to technical challenges, such as low concentration, inefficient extraction, or poor ionization during MS analysis [57]. By incorporating these proteins, even those with barely detectable or missing peptides were accounted for, offering new biological and functional insights—particularly valuable for less-studied species like *L. mutabilis* [58]. Furthermore, the 74 proteins identified with fewer than two unique peptides may still play key roles in cellular functions or metabolic pathways but are often excluded in standard proteomics studies. Examining the full dataset allows researchers to highlight proteins with potential bioactive properties, nutritional significance, or industrial relevance that might otherwise go unnoticed. The classification of these proteins at functional levels 1 and 2 is depicted in Figure 3, with a comprehensive breakdown available in Appendix A.

The distribution of proteins at level 1 was quite similar (Figure 3A) among groups since 201 sequences were classified as cellular components (35.57%), 189 sequences into molecular function (33.45%), and 175 sequences into the biological process group (30.97%). The group of cellular components might reunite the major number of sequences because the primary function of the AF (mainly composed of 2S albumins) is to accumulate in specialized cellular compartments serving as a structural and AA reservoir [59]. In fact, concerning the cellular component group (Figure 3B), most of the proteins were classified within the cellular anatomic structure group (73.89%) followed by the protein-containing complex group (26.10%). These results were in tune with the main functions of albumin-related proteins, as these proteins are primarily localized in protein storage vacuoles (PSVs) which are specialized organelles in seed cells dedicated to storing reserve proteins. Within PSVs, 2S albumins contribute to the formation of protein aggregates or complexes, ensuring the efficient storage and stability of these reserves during seed dormancy [60]. Upon germination, these complexes are mobilized to provide EAAs for the development of seeds [61].

Regarding the biological process group (Figure 3C), the analysis revealed that most of them were related to cellular processes (62.40%), followed by a response to stimulus (19.37%), biological regulations (9.3%), and regulation of biological processes (8.91%). Our results are aligned with previous works about albumin implications in biological processes, as it plays a crucial role in seed viability, growth, and early plant development and its expression may be modulated in response to environmental stresses such as drought or nutrient deficiency [26]. Moreover, albumins have been reported to contribute to the osmotic balance within seed cells by maintaining cellular turgor and stability, which is vital for preserving cell structure during the desiccation and rehydration phases [62]. Finally, within the sequences associated with molecular function at level 2 (Figure 3D), most of them were related to binding (47.03%), followed by catalytic activities (29.26%) and structural molecule activities (23.69%).

### 3.3. Optimization of Albumin Hydrolysis with Protease PC7

In a previous study, the PC7 protease, derived from *Micrococcus* sp. PC7, was isolated and characterized as effective in hydrolysing proteins from different legumes [19]. In the present study, the PC7 protease was concentrated using UF with a 10 kDa MWCO. The volumetric and specific activity of the concentrate protease was 4080 U/mL and 1843.7 U/mg of protein, respectively.

The CCD was applied to optimize both the *DH* and antioxidant activity of HAP by determining the optimal E/S ratio and hydrolysis time. The range of analysis for the E/S ratio was 75–225 U/mg of substrate, and the reaction time was 45–135 min (Table 2). Proteolysis was assessed by the *DH* that was determined by measuring the free amino groups produced during hydrolysis. Values ranging from 18.93% to 39.86% were obtained. The highest values (*DH* > 35%) were reached under conditions with the highest E/S ratios (225–256 U/mg) and longest reaction times (135–153.6 min), as evidenced by treatments 4, 6, and 8. These results could be due to the high enzyme concentration and long incubation times that facilitated the interaction between enzyme and substrate, promoting the bond cleavage. A similar behaviour was observed in the treatments corresponding to the central points (9–11) of E/S (150 U/mg) and times (90 min), achieving a *DH* between 29% and 33%. In contrast, the lowest *DH* values (18.93–26.39%) were associated with conditions involving low E/S ratios (44–75 U/mg) and short reaction times (26.4–45 min), as seen in treatments 1, 3, 5, and 7.

Regarding the antioxidant activity, TEAC values between 139.6 and 235.7 μmol TE/g of protein were obtained. The highest TEAC values (>220 μmol ET/g protein) corresponded to the central points (9–11), reinforcing that moderate E/S conditions and intermediate reaction times favoured the release of antioxidant peptides of adequate size compared to high *DH* treatments (4, 6, and 8).These findings support that high *DH* did not guarantee greater antioxidant activity, as excessive hydrolysis time can lead to the breakdown of BPs into free AAs, thus losing their activity [63]. The antioxidant activity of protein hydrolysates in the ABTS assay is believed to be due to the exposure of specific AAs, such as phenylalanine, tryptophan, tyrosine, valine, and arginine, present in the generated peptides, which interact with the ABTS^•+^ radical, promoting its neutralization and stabilization [9,64].

The SDS-PAGE electrophoretic profile of the 11 runs is presented in Appendix A. The hydrolysates corresponding to treatments 4, 6, and 8 showed a considerable decrease in the protein bands of the FA (10, 14, 20, and 24 kDa) to smaller peptides (<10 kDa), which correlated with the high *DH* achieved (35.24 to 39.86%). However, these treatments failed to gain maximum TEAC values due to prolonged protein hydrolysis, resulting in the release of very short peptides with low antioxidant activity or free AAs [65]. In contrast, treatments 1, 5, and 7 showed remanent FA bands (10–15 kDa), correlated with the low *DH* values achieved (<26%), producing a low amount of antioxidant peptides. In the case of treatments at central points, 9, 10, and 11, a greater intensity of protein bands <10 kDa was evident, related to moderate GH and greater antioxidant capacity (see Table 2).

In summary, a correlation was evident between the protein profile and the *DH* and TEAC results. These results highlight the importance of optimizing the E/S ratio and reaction time since short or excessive protein hydrolysis can limit the production of antioxidant peptides.

In this study, the mathematical models were developed based on the *DH* and antioxidant activity (TEAC) values. These models not only elucidated the relationship between experimental conditions, such as E/S and hydrolysis time, but also allowed the identification of significant interactions between these variables. The mathematical models representing *DH* and TEAC of the optimized HAP were expressed using the following equations:(3)$$DH\left(\%\right)=30.53+5.42X_{1}+13.57X_{2}$$(4)$$TEAC=224.37+26.08X_{1}+39.9X_{2}-37.80X_{1}^{2}-48.82X_{2}^{2}-22.40X_{1}X_{2}$$

The observed values closely matched those predicted by the model (Table 2), confirming the validity of the experimental approach and underscoring the importance of precise parameter control in designing functional processes. The analysis of variance (ANOVA) indicated that *DH* was best represented by a linear model based on its *p*-value, with the E/S ratio variable exhibiting a significant effect (*p* = 0.0118) (Table 3). In contrast, quadratic and interaction effects were not statistically significant (*p* < 0.05). This result suggested that increasing the E/S ratio had a positive effect on *DH* up to a stabilization point, consistent with findings from previous studies [66,67]. On the other hand, the ABTS^•+^ radical scavenging activity of the hydrolysates was best suited to a quadratic model, as indicated by a significant *p*-value (*p* < 0.05). This demonstrated the suitability of the model for optimizing hydrolysis conditions. Additionally, the “lack of fit” values were 0.759 and 0.647 for *DH* and TEAC, respectively, further confirming the adequacy of the models. The coefficient of determination (R^2^), which measured the proportion of explained variation relative to total variation, approached unity for both models (R^2^ = 0.9618 for *DH* and R^2^ = 0.9534 for TEAC), indicating that these regression models accurately represented the system’s behaviour. Similarly, the adjusted R^2^ values (Adj R^2^ = 0.9363 for *DH* and 0.9067 for TEAC) were high, supporting the model’s significance.

The response surface methodology (RSM) provided a graphical representation of data analysis equations and was essential for assessing the influence and interactions among the factors. Figure 4A,B illustrate the contour plots and Figure 4C,D the three-dimensional surface graphs, revealing that the % of *DH* increased progressively toward optimal values, whereas antioxidant activity achieved its maximum value at an E/S ratio of 168 U/mg of substrate and a time of 105 min. To validate the accuracy of the model, a triplicate assay was conducted under the optimal conditions. The obtained values were 32.94 ± 0.53% for *DH* and 227.32 ± 23.78 μmol TE/g of protein for TEAC, showing no significant difference with the predicted values (*p* < 0.05). These findings confirmed that the response model was robust and suitable for predicting the optimization of the hydrolysis reaction.

### 3.4. Hydrolysis of Albumin with Protease PC7 and Alcalase

Sequential hydrolysis was carried out using the PC7 protease and alcalase to enhance the biological activities of HAP [15]. The *DH* increased to 37.88 ± 0.51% after the action of alcalase (Figure 5A), a serine endopeptidase that cleaves peptide bonds at the C-terminal residues of AAs such as glutamic acid, methionine, leucine, tyrosine, lysine, and glutamine [68]. Various *DH* values have been reported for *L. mutabilis* hydrolysates. For example, Chirinos et al. achieved a *DH* of 33.11 ± 2.78% by hydrolysing the protein concentrate with alcalase at 0.385 AU/g protein for 90 min, while the combination with neutrase or flavourzyme did not result in a significant increase in *DH* [21]. Concerning the antioxidant capacity (TEAC value), a significant increase was evident, as shown in Table 4.

Figure 5B illustrates the SDS-PAGE electrophoretic profile of the HAPA hydrolysate, providing a more detailed visualization of the peptide fractions generated during hydrolysis. The increase in *DH* and the reduction of the protein content of HAPA in comparison with HAP demonstrated that AF proteins were effectively degraded by both enzymes. Protein bands above 15 kDa were completely degraded after 60 min of hydrolysis with the PC7, while those below 15 kDa progressively decreased in intensity until the reaction’s completion. The subsequent addition of alcalase facilitated the degradation of residual proteins in the HAP, ultimately producing bands smaller than 4 kDa, corresponding to the generated low MW and antioxidant peptides.

### 3.5. Assessment of Biological Activities

Results revealed that antioxidant activity varied depending on the assay used and increased progressively with hydrolysis time compared to the initial AF (Table 4). The optimized HAP exhibited a 2.7-fold and 2.5-fold increase in TEAC and ORAC values, respectively, compared to unhydrolysed AF. Sequential hydrolysis with PC7–alcalase (HAPA) further enhanced antioxidant activity, achieving a 2.1-fold increase compared to HAP. These findings underscore the beneficial impact of the hydrolysis process on the generation of peptides with potent antioxidant properties, aligning with the results reported by Intiquilla et al. [15]. Similarly, Sbroggio et al. observed a 45% increase in the antioxidant activity of Okara protein hydrolysate compared to the intact protein [69].

Previous studies indicate that the combination of alcalase with other enzymes can promote or decrease antioxidant activity [21,39]. In this study, the synergy between PC7 and alcalase significantly enhanced antioxidant activity, supporting the potential of this enzyme combination as a strategy for functional improvement. The antioxidant activity of protein hydrolysates can be attributed to the presence of low MW peptides enriched with aromatic (such as Tyr, Trp, Phe, and His), hydrophobic (such as Ala, Val, Leu, Pro, and Phe), and acidic (such as Glu and Asp) AAs that accounted for 27.64, 6.54, and 26.42% (Table 1) of the total AAs present, respectively [70,71]. Furthermore, the presence of leucine (3.73 g/100 g) and serine (3.63 g/100 g) in the released peptides has been shown to enhance antioxidant activity, as these amino acids are capable of donating electrons to stabilize the ABTS^•+^ radical [72]. Likewise, the presence of Glu + Glx (20.01 g/100 g) and Arg (8.54 g/100 g) favoured the solubility of these peptides and their antioxidant activity, due to their metal chelating capacity that catalyses the formation of reactive oxygen species (ROS) [73].

The multifunctionality of hydrolysates (HAP and HAPA) was further evaluated through their in vitro antihypertensive and antidiabetic activities by measuring their ACE and DPP-IV inhibitory properties (Appendix A). Regarding the ACE inhibitory capacity, the AF showed an IC_50_ value of 179.6 μg/mL, which decreased significantly (*p* < 0.05) after hydrolysis with PC7 protease (24.9 µg/mL) and the combination PC7 + alcalase (13.5 µg/mL). These results surpassed those reported by Betancour et al., who reported an IC_50_ of 4.5 mg/mL for *Phaseolus vulgaris* protein hydrolysates with the alcalase–flavourzyme combination for 90 min [74]. The enzymatic digestion of *L. albus*, *L. angustifolius*, and *L. luteus* protein fractions with pepsin and trypsin yielded IC_50_ values between 200 and 300 µg/mL [75,76]. The IC_50_ values observed in this study were lower than that reported by Palma-Albino et al. for an AF from *Erythrina edulis* hydrolysed with a sequential combination of pepsin, pancreatin, and alcalase (50.65 μg/mL) [44]. These findings highlighted the variability in ACE inhibitory activity depending on the experimental conditions and the species studied, underscoring the need for optimizing enzymatic digestion processes to maximize the bioactivity of protein hydrolysates. Given the critical role of ACE in blood pressure regulation, particularly its contribution to hypertension, the identification and characterization of ACE inhibitory peptides represent a promising strategy for hypertension prevention.

In the case of the DPP-IV inhibitory activity, the IC_50_ value measured for the non-hydrolysed AF was 1207.9 μg/mL, which was significantly reduced (*p* < 0.05) to 171.3 μg/mL after being hydrolysed with PC7 protease and 145.1 μg/mL when both enzymes were used. The DPP-IV inhibitory activity observed in the present study was higher than that reported in previous research. Chirinos et al. obtained IC_50_ values between 2.14 and 3.21 mg/mL after hydrolysing the Tarwi protein concentrate with alcalase for 120 to 240 min [21]. These values were consistent with the IC_50_ value reported for the optimized protein hydrolysate of *L. albus* (3.42 mg/mL) obtained by sequential hydrolysis with alcalase, trypsin, and flavourzyme [77]. Similarly, the recent study by Muñoz et al., found that the gamma conglutin of *L. mutabilis* did not exhibit DPP-IV inhibitory activity. However, after subjecting the fraction to gastric digestion, it managed to inhibit 100% of DPP-IV at a dose of 5 mg/mL [78].

This study demonstrates the potential of native proteases—used independently or in sequential hydrolysis with commercial enzymes—to produce multifunctional peptides with antioxidant, antihypertensive, and antidiabetic properties. By incorporating such protein hydrolysates into functional foods, the food industry can address critical health challenges, from chronic disease prevention to enhancing the techno-functional and sensory attributes of products. While soy, pea, hemp, and rice proteins dominate the market, rising consumer demand for innovative, health-promoting ingredients creates a compelling opportunity for *L. mutabilis*-derived hydrolysates to gain competitive traction. To fully unlock their commercial and nutritional potential, future research must prioritize the characterization of bioactive peptide fractions, including their precise identification, bioactivity validation, and bioaccessibility—key steps toward advancing functional food and nutraceutical development.

## 4. Conclusions

This study successfully produced a multifunctional protein hydrolysate from the AF of *L. mutabilis* seeds, obtained through TFF, demonstrating its potential as a high-value functional ingredient. The product obtained from the hydrolysis of AF with the PC7 protease under optimal conditions with an E/S ratio of 168 U/mg of substrate and a time of 105 min (HAP) showed high antioxidant activity, as determined by the ORAC method (1.649 μmol TE/g of protein). The multifunctionality of HAP was demonstrated after evaluating its ability to inhibit drug targets associated with hypertension (ACE) and diabetes (DPP-IV). Additionally, sequential hydrolysis of the PC7 protease, followed by alcalase, significantly improved its multifunctional properties (>1.5-fold). HAPA exhibited potent free radical scavenging ability in ABTS and ORAC assays, as well as strong inhibitory activities against ACE (IC_50_ = 13.5 μg/mL) and DPP-IV (IC_50_ = 145.1 μg/mL). These findings highlight the potential use of native PC7 proteases to produce multifunctional protein hydrolysates, potentially useful in the development of functional foods for the prevention of diseases associated with oxidative stress, hypertension, and diabetes. Finally, it is necessary to characterize the bioactive peptides produced in HAP and HAPA, as well as evaluate their bioavailability and bioactivity in biological systems.

## Figures and Tables

**Figure 1 biotech-14-00032-f001:**
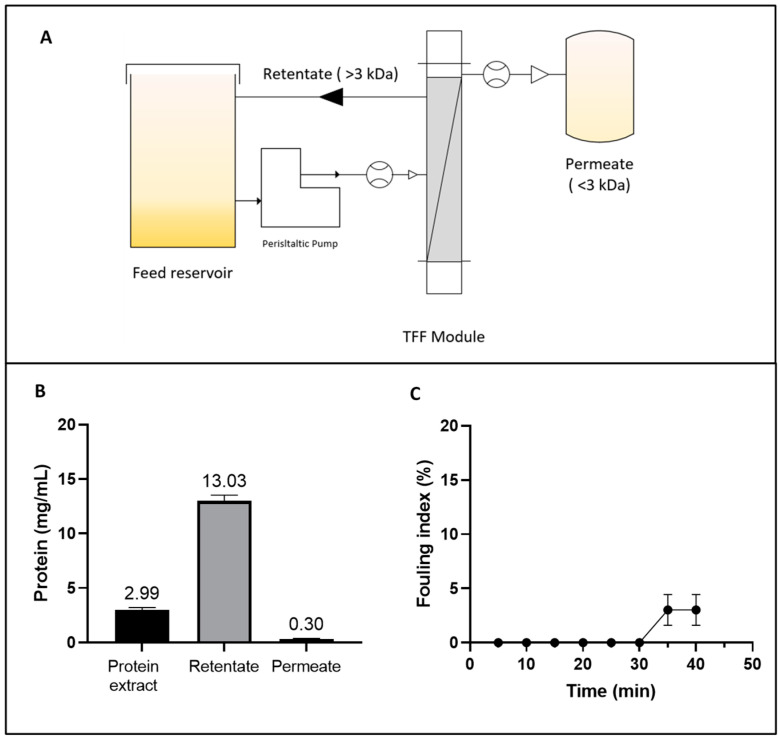
(**A**) Diagram of the tangential flow ultrafiltration (TFF) system for concentrating the albumin fraction (AF) from *Lupinus mutabilis*. (**B**) Protein concentration in the initial extract, retentate, and permeate during the TFF process. (**C**) Fouling index (%) of TFF as a function of time.

**Figure 2 biotech-14-00032-f002:**
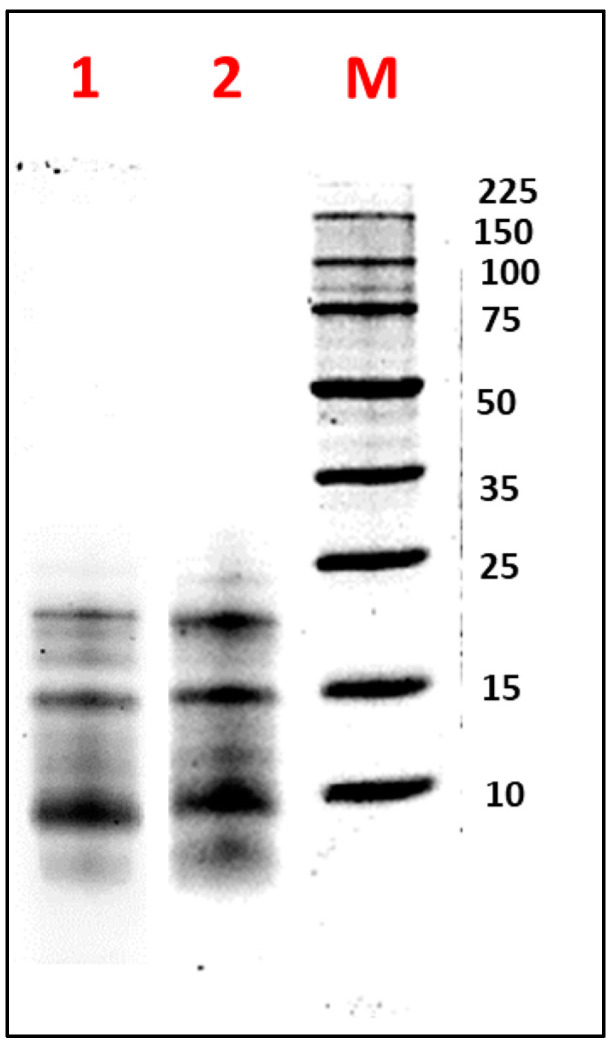
Electrophoretic analysis (SDS-PAGE-Tricine) of the albumin fraction (AF) from *Lupinus mutabilis* “Tarwi” seeds obtained by tangential flow ultrafiltration (TFF), under (1) non-reducing conditions and (2) reducing conditions. M: molecular weight marker (Perfect Protein Marker 10–225 kDa). A total of 10 µg of protein was loaded per sample.

**Figure 3 biotech-14-00032-f003:**
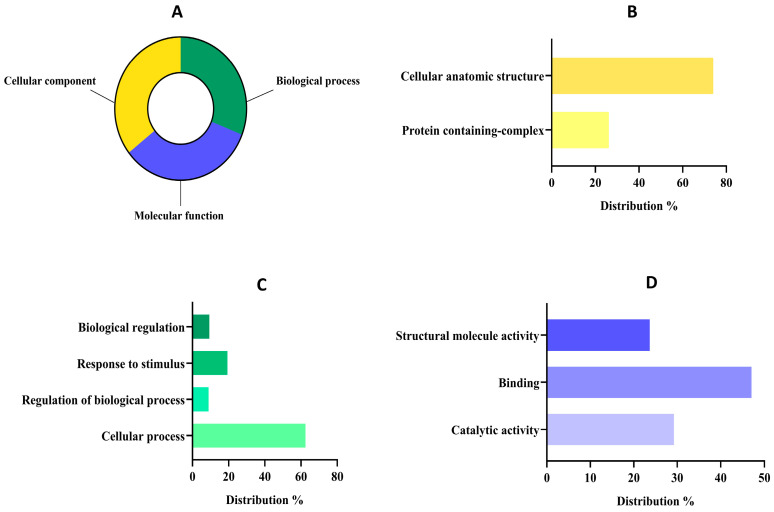
Functional classification of detected proteins from the albumin fraction (AF) of *Lupinus mutabilis* “Tarwi” based on gene ontology (GO). (**A**) Proteomic percentage distribution at GO level 1 classification. (**B**) Molecular function GO level 2 distribution. (**C**) Biological process GO level 2 distribution. (**D**) Cellular component GO level 2 distribution.

**Figure 4 biotech-14-00032-f004:**
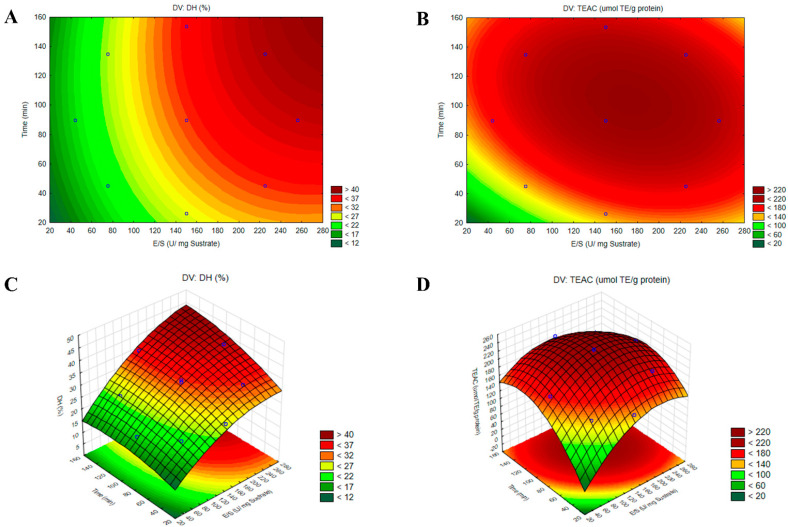
Contour plots (**A**,**B**) and three-dimensional surface plots (**C**,**D**) illustrating the effects of the enzyme-to-substrate ratio (E/S) and hydrolysis time (t) on the responses for the degree of hydrolysis (*DH*%) and ABTS^•+^ radical antioxidant activity (TEAC) in the optimized hydrolysate from the albumin fraction (AF) of *Lupinus mutabilis* with protease PC7 (HAP).

**Figure 5 biotech-14-00032-f005:**
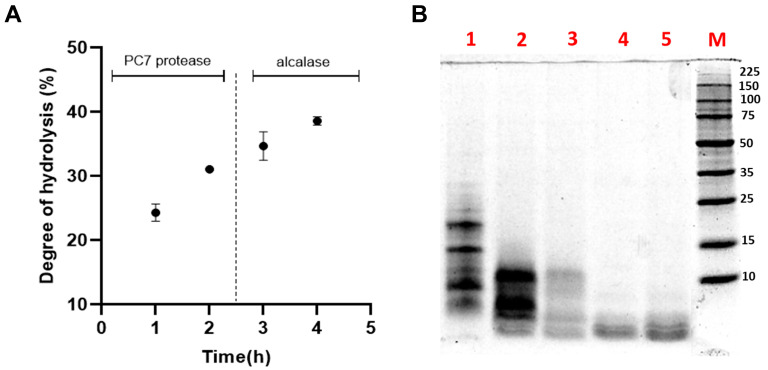
(**A**) Effect of sequential hydrolysis of the albumin fraction (AF) from *Lupinus mutabilis* albumin using protease PC7 and alcalase on the (**A**) degree of hydrolysis (*DH*); (**B**) protein profile analysed by SDS-PAGE under reducing conditions. Lane 1, non-hydrolysed AF; lanes 2 and 3, hydrolysis of AF with protease PC7 for 1 and 2 h, respectively; and lanes 4 and 5, hydrolysis of AF with alcalase for 1 and 2 h, respectively. M: molecular weight marker (Perfect Protein Marker 10-225 kDa, Sigma-Aldrich).

**Table 1 biotech-14-00032-t001:** Amino acid composition (g/100 g protein and g/100 g biomass) of albumin fraction (AF) from *Lupinus mutabilis*.

Amino Acid	AF from *L. mutabilis*	
	g/100 g Protein	g/100 g Biomass
Essential (EAA)		
Lysine (Lys)	3.52 ± 0.05	2.92 ± 0.04
Phenylalanine (Phe)	2.38 ± 0.03	1.97 ± 0.03
Tyrosine (Tyr)	2.06 ± 0.11	1.71 ± 0.09
Methionine (Met)	0.34 ± 0.03	0.28 ± 0.03
Cysteine (Cys)	1.42 ± 0.12	1.17 ± 0.10
Threonine (Thr)	2.38 ± 0.04	1.97 ± 0.03
Leucine (Leu)	3.73 ± 0.02	3.09 ± 0.02
Isoleucine (Ile)	1.91 ± 0.07	1.58 ± 0.06
Valine (Val)	1.70 ± 0.04	1.41 ± 0.03
Non-essential (NEAA)		
Aspartic acid + Asparagine (Asx)	6.41 ± 0.03	5.31 ± 0.03
Glutamic acid + Glutamine (Glx)	20.01 ± 0.26	16.58 ± 0.21
Serine (Ser)	3.63 ± 0.06	3.01 ± 0.05
Histidine (Hys)	2.07 ± 0.05	1.72 ± 0.04
Arginine (Arg)	8.54 ± 0.29	7.08 ± 0.24
Alanine (Ala)	2.31 ± 0.02	1.91 ± 0.01
Proline (Pro)	2.91 ± 0.00	2.41 ± 0.00
Glycine (Gly)	2.55 ± 0.02	2.11 ± 0.02
EAA	19.44	16.10
NEAA	48.43	40.13
TAA	67.87	56.23
EAA × 100/TAA (%)	28.64	
EAA × 100/NEAA (%)	40.14	
HAA × 100/TAA (%)	27.64	
AAA × 100/TAA (%)	6.54	

EAA: essential amino acids; NEAA: non-essential amino acids; TAA: total amino acids; HAA: hydrophobic amino acids (Ala + Val + Ile + Leu + Tyr + Phe + Trp + Met + Pro + Cys); AAA: aromatic amino acids (Phe + Trp + Tyr).

**Table 2 biotech-14-00032-t002:** Observed and predicted values for degree of hydrolysis (*DH*%) and ABTS^•+^ radical scavenging activity (TEAC) from the experimental design optimizing enzyme/substrate ratio (E/S) and hydrolysis time (T) of the albumin fraction (AF) from *Lupinus mutabilis* hydrolysed with protease PC7 (HAP).

Run	X1	X2	Y1		Y2	
	E/S	Time	*DH* (%)		TEAC (µmol TE/g Protein)	
	(U/mg)	(min)	Observed	Predicted	Observed	Predicted
1	75	45	22.57 ± 0.35	18.94	142.3 ± 7.26	136.9
2	225	45	32.91 ± 0.74	23.55	193.3 ± 0.19	185.4
3	75	135	25.14 ± 0.93	29.40	195.0 ± 9.74	199.1
4	225	135	39.49 ± 0.88	34.01	201.2 ± 9.52	202.8
5	150	26.4	26.39 ± 0.69	20.03	139.6 ± 2.86	143.7
6	150	153.6	35.24 ± 1.26	34.82	208.7 ± 10.00	203.8
7	44	90	18.93 ± 0.45	22.28	168.0 ± 8.44	168.1
8	256	90	39.86 ± 1.02	28.79	201.3 ± 14.54	205.0
9	150	90	33.38 ± 2.35	28.29	224.6 ± 12.50	224.3
10	150	90	32.54 ± 1.85	28.29	222.8 ± 11.08	224.3
11	150	90	29.40 ± 2.15	28.29	235.7 ± 7.06	224.3

**Table 3 biotech-14-00032-t003:** Analysis of variance (ANOVA) for the quadratic response surface model for the degree of hydrolysis (*DH*%) and ABTS^•+^ radical antioxidant activity (TEAC) of the optimized hydrolysate from the albumin fraction (F) of *Lupinus mutabilis* with protease PC7 (HAP).

Cause	Sum of Squares	Df	Middle Square	Value *f*	*p*
*DH* (%)					
(1) Time (L)	58.6758	1	58.6758	13.30566	0.067622
Time (Q)	1.3973	1	1.3973	0.31687	0.630181
(2) E/S (L)	368.5564	1	368.5564	83.57598	0.011755
E/S (Q)	8.2223	1	8.2223	1.86453	0.305398
Lack of Fit	8.4929	4	2.1232	0.48147	0.759349
Pure error	8.8197	2	4.4098		
R^2^	0.9618				
R^2^ adjusted	0.9363				
TEAC (μmol TE/g protein)					
(1) E/S (L)	1360.623	1	1360.623	11.97859	0.074298
E/S (Q)	2017.619	1	2017.619	17.76261	0.050195
(2) Time (L)	3187.316	1	3187.316	28.06033	0.033839
Time (Q)	3364.309	1	3364.309	29.61854	0.032144
1L by 2L	501.975	1	501.975	4.41927	0.170278
Lack of Fit	225.913	3	75.304	0.66296	0.647924
Pure error	227.176	2	113.588		
R^2^	0.9534				
R^2^ adjusted	0.9067				

**Table 4 biotech-14-00032-t004:** Antioxidant activity (expressed as µmol TE/g for ABTS^•+^ and µmol TE/mg protein for ORAC), antihypertensive activity (expressed as IC_50_ for ACE inhibition), and antidiabetic activity (expressed in IC_50_ for DPP-IV inhibition) of the *Lupinus mutabilis* albumin fraction and its enzymatic hydrolysates (HAP and HAPA).

Sample	ABTS	ORAC	ACE	DPP-IV
	µmol TE/g Protein	µmol TE/g Protein	IC_50_ (µg/mL)	IC_50_ (µg/mL)
Albumin fraction (AF)	92.0 ± 2.9 ^a^	646.7 ± 55.3 ^a^	179.6 ± 5.5 ^a^	1207.9 ± 56.7 ^a^
HAP	251.1 ± 11.0 ^b^	1649.0 ± 75.7 ^b^	24.9 ± 0.9 ^b^	171.3 ± 10.3 ^b^
HAPA	536.7 ± 6.8 ^c^	3193.3 ± 230.3 ^c^	13.5 ± 1.1 ^c^	145.1 ± 6.8 ^c^

^a,b,c^ Different letters in the same column represent significant differences (*p* < 0.05).

## Data Availability

The original contributions presented in the study are included in the article/Appendix A. Mass spectrometry proteomics data are available through ProteomeXchange under the identifier PXD062562; further inquiries can be directed to the corresponding authors.

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
