# Peer review of "Production of Multifunctional Hydrolysates from the Lupinus mutabilis Protein Using a Micrococcus sp. PC7 Protease"

_biotech, 2025, doi:10.3390/biotech14020032_

Round 1
Reviewer 1 Report
Comments and Suggestions for Authors
Comments to the authors:
In this paper, the antioxidant activity and related enzyme inhibitory activity of hydrolysates from Lupinus mutabilis protein were systematically studied, which provided valuable experimental data and sufficient results analysis. However, there are some corrections that need to be made to the presentation of the data and the use of some terms. The following are the specific review comments and suggestions:
- Line 279: The abbreviation “TE/mg” should be corrected for clarity and consistency.
- Line 355: The term “The volumetric flow rate (LMH)” should be accurately presented. Please ensure the correct terminology is used.
- LC-chromatogram of amino acids: The authors should provide the LC-chromatogram for the quantification of amino acids to support their data.
- Table 1: The formula “EAA100/TAA (%) - AAA100/TAA (%)” should be corrected for accuracy.
- LC-chromatogram of albumin fraction (AF): The authors should include the LC-chromatogram for the identification of the albumin fraction (AF) sample.
- Table 2: The notation “TEAC (µmol ET/g protein)” should be presented correctly.
- Line 498: The reference numbers “(3, 4 and 6)” should be corrected to “(4, 6, 8)” for consistency.
- Line 506: The abbreviation “GH” should be corrected to “DH” for accuracy.
- Line 560: The increase in DH from 31.07 ± 0.43% to 37.88 ± 0.51% is not clearly reflected in Figure 5A. Please correct this discrepancy.
- Table 3: The notation “TEAC (µmol/g protein)” should be presented correctly.
- Line 576: The abbreviation “GH” should be corrected to “DH” for accuracy.
- Table 4: The notation “TEAC (µmol ET/g protein)” should be presented correctly.
- Table 4: The abbreviation “ECA” should be corrected to “ACE” for consistency.
- Table 4: The symbol “c” is unclear. Please provide a clear definition or explanation for completeness.
- Antioxidant and enzyme activities: Regarding the activities of ABTS, ORAC, ACE, and DPP-IV, it is recommended that the authors provide inhibition data graphs with concentration gradients to better illustrate their findings.
- Line 669: Please confirm the basis for the data of “TEAC value of 1.649 µmol ET/mg protein.”
- Formatting: In the manuscript, please unify the format to (p < 0.05) for consistency.
Author Response
"Please see the attachment."

Reviewer 2 Report
Comments and Suggestions for Authors
Production of multifunctional hydrolysates from Lupinus Mutabilis protein using a Micrococcussp. PC7 protease
Major: The author is suggested to improve the English language
Abstract: The author is suggested to improve the quality of abstract
Introduction:
- Line 40, The author is suggested to give clarity that how anti-nutritional factors can be bioactive components
- Line 46-47, the author is suggested to remove the very generalised information on bioactive peptides
- Line 60, the author is suggested to give potential applications of pepsin, pancreatin, flavorzyme, neutrase, and alcalase in human health.
- Line 66, the author is suggested to give clarity on what are mild conditions and what does it mean exactly about avoiding of high temperatures.
- Line 73-74, the author is suggested to remove the generalised information
- Line 71-80, the authors are suggested to give very specific information rather than generalised information
Methodology:
- Line 240, the author is suggested to give % DH in the equation format
- 272, how did the reaction mixture was protected from light, the author is suggested for the clarification
- The author is suggested to include the reference of 2.8.4. Dipeptidyl peptidase-IV (DPP-IV) inhibitory activity
- The author is suggested to give reference for the Sequential hydrolysis of albumin fraction with Protease PC7 and Alcalase (HAPA)
- The author is suggested to write clearly about 2.9. Statistical analysis
Results:
- Line 326, The author is suggested replace sulphur-rich protein to sulphur rich protein constituent
- Line 371, the author is suggested to give clarification on why Trp is not detected during the process of acid hydrolysis.
- Line 382, what is the differences between the non reducing and reducing conditions, the author is suggested for the clarification
- Line 416, the author is suggested to give clarity on what does it mean 242 proteins.
- Line 422, the author is suggested to give clarity on what does it mean 74 proteins
- The author is suggested for the references in the 451 and 452
- Line 465-466, the author is suggested to remove the generalised procedure that they can discuss in the methodology section
- The author is suggested to give appropriate references for 505-519
- The author is suggested to replace the old references with the newer ones
- The author is suggested to recheck the conclusion part
Author Response
"Please see the attachment."

Reviewer 3 Report
Comments and Suggestions for Authors
Dear Authors,
The topic covered is relevant to the field of biotechnology, and the general structure is well outlined. However, I believe some revisions are necessary to improve the scientific quality and impact of the work. I would like to make some suggestions below to assist you in this matter:
- As it stands, it is not sufficiently clear what is new about your study compared to the existing literature. I recommend that you explicitly highlight the original contribution in the introduction (see lines 90-98).
- Some of the conclusions drawn do not seem to be fully supported by the data presented. I suggest a revision of the discussion section to strengthen the internal logic and emphasize how the results justify the final claims (see lines 674-679).
- I recommend a careful revision of the text to correct ambiguous wording and improve the coherence of the paragraphs. In particular, some transitions between ideas could be made smoother (see lines 500-519). The SDS-PAGE should be described where the corresponding figure is mentioned, and each analysis should be discussed where it is mentioned and not always referred to.
- I believe that the discussion could benefit from an additional or expanded section dealing with practical implications or possible applications of the results. This could help to emphasise the relevance of the study. Please also check that the references are written in MDPI format (see lines 547-556). Discussions about figures do not always take place where they are first
- The text contains some inconsistent wording or repetitions that can be corrected by careful reading. A final linguistic check would be advantageous before a possible republication.
I hope that these suggestions will be helpful in the revision process.
Round 2
Reviewer 2 Report
Comments and Suggestions for Authors
The quality of the manuscript is now improved. In my opinion, this version can be accepted for publication consideration.